# CELF2 Sustains a Proliferating/OLIG2+ Glioblastoma Cell Phenotype via the Epigenetic Repression of SOX3

**DOI:** 10.3390/cancers15205038

**Published:** 2023-10-18

**Authors:** Laurent Turchi, Nathalie Sakakini, Gaelle Saviane, Béatrice Polo, Mirca Saras Saurty-Seerunghen, Mathieu Gabut, Corine Auge Gouillou, Vincent Guerlais, Claude Pasquier, Marie Luce Vignais, Fabien Almairac, Hervé Chneiweiss, Marie-Pierre Junier, Fanny Burel-Vandenbos, Thierry Virolle

**Affiliations:** 1CNRS, INSERM, Institut de Biologie Valrose, Team INSERM “Cancer Stem Cell Plasticity and Functional Intra-tumor Heterogeneity”, Université Côte D’Azur, 06107 Nice, France; laurent.turchi@unice.fr (L.T.); nvs30@cam.ac.uk (N.S.); saviane.gaelle@hotmail.fr (G.S.); beatrice.polo@univ-cotedazur.fr (B.P.); almairac.f@chu-nice.fr (F.A.); burel-vandenbos.f@chu-nice.fr (F.B.-V.); 2DRCI, CHU de Nice, 06107 Nice, France; 3CNRS UMR8246, INSERM U1130, Neuroscience Paris Seine-IBPS Laboratory, Team Glial Plasticity and NeuroOncology, Sorbonne Université, 75252 Paris, France; sarasmsaurty@gmail.com (M.S.S.-S.); herve.chneiweiss@inserm.fr (H.C.); marie-pierre.junier@inserm.fr (M.-P.J.); 4Stemness in Gliomas Laboratory, Cancer Initiation and Tumoral Cell Identity (CITI) Department, INSERM 1052, CNRS 5286, Centre Léon Bérard, 69008 Lyon, France; mathieu.gabut@inserm.fr; 5Cancer Research Center of Lyon 1, Université Claude Bernard Lyon 1, 69100 Villeurbanne, France; 6UMR 1253, iBrain, Inserm, Université de Tours, 37000 Tours, France; auge@univ-tours.fr; 7CNRS, I3S, Université Côte d’Azur, 06560 Valbonne, France; guerlais@i3s.unice.fr (V.G.); claude.pasquier@unice.fr (C.P.); 8CNRS, INSERM, Institut de Génomique Fonctionnelle, IGF, Université de Montpellier, 34090 Montpellier, France; marie-luce.vignais@igf.cnrs.fr; 9Service de Neurochirurgie, Hôpital Pasteur, CHU de Nice, 06107 Nice, France; 10Service d’Anatomopathologie, Hôpital Pasteur, CHU de Nice, 06107 Nice, France

**Keywords:** glioblastoma, cancer stem cells, epigenetic, RNA binding protein, H3K9me3, OLIG2, CELF2

## Abstract

**Simple Summary:**

Glioblastomas, primitive infiltrating brain tumors, are a real public health problem because of their dismal prognosis. The persistence of aggressive tumor stem cells after conventional cytotoxic treatment is one of the major causes of therapeutic failure. The identification of targets involved in the molecular mechanisms repressing the aggressive stem cell phenotype is a relevant approach that offers an alternative to more conventional cytotoxic therapies, which have failed to prevent GBM recurrence. In this study, we have identified the protein CELF2 as being an efficient epigenetic regulator of genes in glioma stem cells (GSCs). CELF2 shapes a H3K9me3-repressive landscape in the SOX3 gene, thereby promoting a proliferating tumor cell phenotype. We found that CELF2 is a major point of tumor vulnerability as its repression is sufficient to convert aggressive tumor cells into cells without the ability to form tumors in vivo. CELF2 is a crucial target that warrants attention for the development of novel anticancer strategies.

**Abstract:**

Glioblastomas (GBs) are incurable brain tumors. The persistence of aggressive stem-like tumor cells after cytotoxic treatments compromises therapeutic efficacy, leading to GBM recurrence. Forcing the GBM cells to irreversibly abandon their aggressive stem-like phenotype may offer an alternative to conventional cytotoxic treatments. Here, we show that the RNA binding protein CELF2 is strongly expressed in mitotic and OLIG2-positive GBM cells, while it is downregulated in differentiated and non-mitotic cells by miR-199a-3p, exemplifying GBM intra-tumor heterogeneity. Using patient-derived cells and human GBM samples, we demonstrate that CELF2 plays a key role in maintaining the proliferative/OLIG2 cell phenotype with clonal and tumorigenic properties. Indeed, we show that CELF2 deficiency in patient-derived GSCs drastically reduced tumor growth in the brains of nude mice. We further show that CELF2 promotes TRIM28 and G9a expression, which drive a H3K9me3 epigenetic profile responsible for the silencing of the SOX3 gene. Thus, CELF2, which is positively correlated with OLIG2 and Ki67 expression in human GBM samples, is inversely correlated with SOX3 and miR-199a-3p. Accordingly, the invalidation of SOX3 in CELF2-deficient patient-derived cells rescued proliferation and OLIG2 expression. Finally, patients expressing SOX3 above the median level of expression tend to have a longer life expectancy. CELF2 is therefore a crucial target for the malignant potential of GBM and warrants attention when developing novel anticancer strategies.

## 1. Introduction

Tumor development involves mutation accumulation, epigenetic modifications, and interactions with reactive non-malignant cells residing in the tumor microenvironment. This process leads to dynamic changes in tumor cell phenotype and behavior, resulting in the development of a heterogeneous biotope. In recent years, this intra-tumor heterogeneity has been particularly explored in glioblastoma (GB), the most aggressive form of primary brain tumor. This tumor, characterized by multiple tumor cell populations harboring distinct genomic abnormalities, as well as by epigenetic and transcriptomic signatures [1,2,3,4] is a paradigm of cellular and molecular heterogeneity. For example, cells that harbor heterogeneous genomic abnormalities, such as EGFR and PDGFR amplifications or different transcriptomic patterns, originally thought to characterize subclasses of GBs, have been identified within the same tumor [1,4]. Further genetic, genomic, and single-cell histological studies have confirmed the high levels of cellular heterogeneity by identifying distinct tumor cell populations in varying proportions, with dissimilar morphologies, genetic characteristics, and proliferating capacities [5,6,7]. Of note, cells that possess divergent capabilities while also sharing similar genomic abnormalities have been identified [8]. Recent single-cell transcriptomic studies have revealed similarities between the transcriptomic profiles of GBs and normal cells [9,10], delineating distinct general cellular identities or states, such as neural precursors (NPC-like), oligodendrocyte precursors (OPC-like), mesenchymal (MES-like), and astrocytes (AC-like), as well as smaller populations exhibiting hybrid states [9,10]. The authors reported that GBM cells could switch between malignant cellular states, with each contributing to tumor growth. These data reinforce the idea that GBM cells are highly plastic cells, a view emphasized by the results of previous cellular biology studies that have identified signaling pathways and epigenetic regulations as mediating the transitions of GBM cells between highly aggressive and more indolent phenotypes [8,11,12,13,14]. In this context, we demonstrated that indolent differentiated GBM cells feed aggressive tumor cell populations by spontaneously reprogramming into OLIG2-, NANOG-, and OCT4-positive cells with stem-like and tumorigenic properties [11,13]. The persistence of aggressive GBM cells after conventional treatments leads to systematic tumor relapse and a dramatically short median of survival of 18 months [15,16]. Thus, targeting key trans-acting factors, which control the maintenance of aggressive tumor cell phenotype, would be a relevant strategy for establishing new therapies against GBMs.

CELF2 is a member of the CELF/Bruno-like family of RNA binding proteins [17]. Localized in the nucleus and cytoplasm, CELF2 is involved in molecular processes that are specific to either compartments, such as alternative splicing in the nucleus or translational regulation in the cytoplasm [17,18,19]. Therefore, CELF2 is a major player in gene expression and regulation. So far, convincing studies in various cancers, including gliomas, have shown that CELF2 plays the role of tumor suppressor. According to these reports, the targeting of CELF2 by several oncogenic microRNAs or its repression via promoter methylation causes its underexpression in tumors compared to normal tissues. In this context, the restoration of CELF2 expression leads to impaired cell proliferation and migration and stimulates sensitivity to apoptosis [20,21,22,23,24]. Accordingly, these studies have associated CELF2 expression with a better prognosis of the disease [20,21,23,25]. However, nothing has yet been reported on CELF2 expression and function in the context of intra-tumoral glioma heterogeneity and, more specifically, in patient-derived glioma stem cells (GSCs) and their differentiated counterparts. In the present study, we explored the expression profile and mode of action of CELF2 in our collection of GBM tissue samples representative of mitotic tumor territories enriched in OLIG2-positive cells or, conversely, non-mitotic tumor areas poor in OLIG2-positive cells [8,13]. To this end, we also used, 3D cultures of patient-derived cells (PDCs), maintained as self-renewing GSCs spheroids, in serum-free media [8,13]. Surprisingly, we found that CELF2 correlates with OLIG2 expression and confers a malignant potential to GSCs.

## 2. Materials and Methods

### 2.1. Cell Culture

PDCs were isolated from surgical resections of primary human GBMs supplied by the Department of Neurosurgery of the University Hospital of Nice (GB1, GB5, GB11). TG6 cells were supplied by Hervé Chneiweiss, Université Pierre et Marie Curie, Paris. Tumor cell dissociation was performed as described elsewhere [26]. All our PDC cultures have been enriched in GSCs by subsequent rounds of clonal amplification. The spheroids were cultured in NS34+ medium containing EGF and bFGF (DMEM-F12 1:1 ratio, 10 mM glutamine, 10 mM HEPES, 0.025% sodium bicarbonate, N2, G5, and B27 supplements). Differentiation experiments were performed in serum-containing media (DMEM-F12, 10 mM Glutamine, 10 mM Hepes, 0.025% sodium bicarbonate (*w*/*v*), 1% FCS (*v*/*v*)). GB5 and TG6 display a gain on chromosome 7, without the loss of PTEN, while GB1 displays a gain on chromosome 7 with the loss of the whole chromosome 10. GB1, GB5, and TG6 do not show mutations of IDH1/2 and TP53, active SHH, and NOTCH pathways, respectively.

Where indicated, cells were treated with G9a-IN-1 (15 µM) (Clinisciences, 74, rue des Suisses 92000 Nanterre, France).

Where indicated, cells were transfected using Lipofectamine 2000 reagent (Life Technologies, Courtaboeuf, France) with miR199a-3p (MC11779), miR199a-5p (MC10893), miR214 (MC12921), miR30a (MC11062), Anti-miR199a-3p (AM11779), a miRNA negative control (Scramble-4464061), a silent siRNA directed against SOX3 (115698 and 45128), or a silent negative control (4404021) (Life Technologies, Courtaboeuf, France) at 100 nM for 72 h. Cells were then lysed to collect total RNA or protein or fixed with paraformaldehyde (4% *w*/*v*) for experiments.

### 2.2. siRNAs and shRNAs

The following siRNAs and shRNAs were used.

SOX3-specific siRNA: Reference siRNA ID115698: siSOX3-1; siRNA ID16562: siSOX3-3, (Thermofisher Scientific, Saint-Herblain, France).

CELF2-specific shRNA: HSH000802-1 shCELF2-1: 5′-ccgcagagtaaaggttgtt-3′, HSH000802-2 shCELF2-2: 5′-ggcatgaatgctttacagt-3′, HSH000802-3 shCELF2-3: 5′-gctatccaagctatgaatg-3′, HSH000802-4 shCELF2-4: 5′-gacagcaaaccttactgat-3′. HSH000802-2 shCELF2-2 has been used to provide shCELF2 GB5. HSH000802-3 shCELF2-3 has been used to provide shCELF2 TG6 and GB1 cells.

Scrambled shRNA and scrambled siRNA were used as controls.

### 2.3. Orthotopic Xenografts

Cells of 5 × 10^4^ from GB5shCTL (*n* = 5), GB5shCELF2 (*n* = 5), GB1shCTL (*n* = 5), and GB1 shCELF2 (*n* = 5), engineered to stably express a luciferase reporter gene as previously described, were resuspended in 5 μL of Hanks’ balanced salt solution (Invitrogen, Fisher Scientific, Illkirch, France) for stereotactic implantation in the right striatum of male nude mice (Janvier Labs, Le Genest-Saint-Isle France). Cell survival and tumor growth were monitored and quantified in living animals up to 90 days after transplantation using the IVIS Lumina system (Caliper Life Sciences, Hopkinton, MA, USA).

### 2.4. RNA Sequencing Analysis

Total RNAs were extracted from GB5-shCELF2 and GB5-shCTL and supplied to the Nice-Sophia-Antipolis functional genomics platform. The samples were sequenced on the Illumina NextSeq500. The sequence libraries (reads) obtained were aligned with STAR to the hg19 genome version in the primary analysis. Differences in relative library size between samples are normalized by calculating size factors for each sample with the DESeq2 package. Size factors were calculated by constructing a reference value for each gene defined by its geometric mean count over all samples. Then, for each gene, the count values in each sample were normalized (divided) by the previously calculated reference value. For each sample, an average value of the normalized values of all genes was calculated and this average value represents the size factors per sample. Differential analysis is performed for the following contrasts: shCELF2 versus shCTL. “DESeq2” package is used to perform the statistical modeling.

### 2.5. ChIP Sequencing (ChIP-Seq) and Bioinformatics Analysis

Cells were treated with a 1% formaldehyde cross-linking solution for 10 min at room temperature. The cross-linking reaction was stopped by adding 125 mM glycine for 5 min at room temperature. After three washes with cold PBS, cells were dry-frozen and sent to Active Motif for H3K9me3 immunoprecipitation and library sequencing. Briefly, 75 nt sequence reads were generated by Nextseq500 (Illumina, Paris, France). Downstream bioinformatics analysis was performed using Active Motif using BWA software for the Hg38 human genome version. Peak calling was performed with MACS and SICER software.

### 2.6. Spheroid Formation Assays

Cells were seeded at 1 cell/well in 96-well plates for each condition. The number of wells containing a spheroid was counted 3 weeks after seeding. Experiments were repeated three times independently. At least 200 wells were counted. The *p* values were calculated using Student’s *t*-test.

### 2.7. Western Blot Analysis

Proteins were extracted at the indicated time with lysis buffer (50 mM Tris-HCl pH7.6, 150 mM NaCl, 5 mM EDTA, 1% NP40). After migration, proteins were transferred to Immobilon P membranes (Millipore Corporation, Bedford, MA, USA) and probed with antibodies (1 μg/mL) to CELF2 (#HPA035813, MerckMillipore, Saint-Quentin-Fallavier, France), EHMT2 (#sc-515726, Santa Cruz Biotechnology, Heidelberg, Germany), anti-KAP1 (#ab109545, Abcam, Paris, France), SOX3 (#sc-101155, Santa Cruz, Biotechnology, Heidelberg, Germany), H3K9me3 (#ab8898, Abcam, Paris, France), ß-Tubulin (#MA5-16308, ThermoFisher, Saint-Herblain, France), or ERK1 (#sc94-Santa-Cruz Biotechnology, Heidelberg, Germany) overnight at 4 °C. Western blots were then washed with TBS-T buffer (Tris 20 mM pH7.4, NaCl 150 mM, Tween-20 0.5%), and probed with HRP-conjugated antibodies. Protein expression levels were assessed with ECL (Clarity, Biorad, Marnes-la-Coquette, France) using a Biorad imager. Quantifications were performed using Image J 2.0.0 software.

### 2.8. Immunofluorescence

Cells were seeded on poly-lysine-coated glass slides in NS34+ medium or serum. At indicated times, cells were fixed with 4% paraformaldehyde (*w*/*v*) for 15 min at room temperature. Blocking and hybridization were performed in PBS containing 10% FCS and 0.1% Triton X100 with the primary antibodies as indicated in the Appendix A. Washes were performed in PBS. Secondary antibodies used were coupled to Alexa-488, Alexa-548, Alexa-647, and nuclei were stained with Hoechst 33342. Immunofluorescences were imaged with a TI-Eclipse microscope, captured with a Hamamatsu camera and acquired and quantified with NIS-Elements software (Nikon Instrument Inc., New York, NY, USA).

### 2.9. Immunohistochemistry and Immunohistofluorescence

Deparaffinization, rehydration, and antigen retrieval were performed using the PTlink pretreatment module (Dako, Les Ulis, France). Immunostaining was performed using a DAKO automaton or manually with the following primary antibodies: anti-CELF2, anti-Olig2, anti-MIB-1, anti-CCNA, anti-SOX3—as indicated in the Appendix A. Immunostaining was scored independently by a pathologist (F.B.V.) and a researcher (T.V.).

### 2.10. RNA Immunoprecipitation PCR Assay

RNA and protein complexes of GB5shCTL and GB5shCELF2 were cross-linked via UVC at 250 J/m^2^ with a stratalinker irradiator. RIP experiments were carried out following the protocol described previously [27]. Irradiated cells were washed three times with ice-cold PBS and then resuspended in polysome lysis buffer (100 mM KCl, 5 mM MgCl_2_, 10 mM HEPES, 0.5% NP40, 1 mM DTT, 100 units/mL RNase Out, 400 μM Vanadyl Ribonucleoside Complex (VRC), protease and phosphatase inhibitor cocktail (Thermofischer Scientific)). The resulting mRNP lysates were homogenized with Dounce, incubated on ice for 5 min and stored at −80 °C. Thawed mRNP lysates were centrifuged at 15,000× *g* for 15 min at 4 °C and the cleared supernatants were diluted 10-fold in NT2 buffer (50 mM Tris-HCl (pH 7.4), 150 mM NaCl, 1 mM MgCl_2_, 0.05% NP40, 100 units/mL RNase Out, 400 uM Vanadyl Ribonucleoside Complex (VRC), protease and phosphatase inhibitor cocktail). The A/G protein-coated CELF2 antibody (clone 1H2, Sigma-Aldrich, Saint-Quentin-Fallavier, France) was incubated with the mRNP lysates overnight at 4 °C with constant shaking. The beads were washed 5 times with NT2 buffer, and then RNA was isolated with Trizol reagent (Thermofisher Scientific, Saint-Herblain, France). RNAs were reverse transcribed with the High-Capacity Reverse Transcription Kit (Applied Biosystem, Thermofisher Scientific, Saint-Herblain, France). Real-time qPCR for EHMT2 and TRIM28 was performed using the SybrGreen master mix (Applied Biosystem, Thermofisher Scientific, Saint-Herblain, France) with the specific primers for EHMT2 and TRIM28 (see Appendix A). CT values were normalized to the CT values detected via GAPDH in all samples. Fold enrichment was calculated by the 2^−∆CT^ method between CTs obtained in knockdown vs. control conditions.

### 2.11. Statistical Analysis

The significance of the observed differences was determined via Student’s *t* test. Spearman’s correlation coefficient was used to calculate the correlations presented in the study. The Wilcoxon Mann–Whitney test was used to estimate the significance of differences in tumor development. Kaplan–Meier analysis was used for survival analysis, and differences in survival probabilities were estimated using the log-rank test; *p* < 0.05 was considered to indicate statistical significance. Statistical analyses were performed using Biostatgv (https://biostatgv.sentiweb.fr, INSERM and Sorbonne University, Paris, France, URL accessed on 22 July 2023).

## 3. Results

### 3.1. Characterization of CELF2 Expression in GBM Samples

We first sought to analyze CELF2 protein expression in paraffin-embedded GBM tissue sections. Immunostaining, using a CELF2-specific antibody, revealed a higher percentage of CELF2+ cells in tumor areas with a high cell density, compared to areas with a lower cell density (mean 77% +/− 13 versus 18% +/− 5 CELF2+ cells), suggesting that CELF2 may be involved in the regulation of cell proliferation and tumor growth (Figure 1A). This hypothesis was evaluated by determining the correlation index between CELF2+ and OLIG2+/Ki67+ cells in GBM, using a collection of 20 GBM samples, 10 corresponding to a tumor zone enriched with Ki67 and OLIG2-positive cells, and 10 exemplifying a GBM zone with a low mitotic index and impoverished in OLIG2-positive cells [8]. Markers’ expression on consecutive GBM slices have shown that CELF2+ cells were more frequently associated with Ki67+/OLIG2+ territories (mean 79% +/− 12) than with Ki67-/OLIG2- territories (11% +/− 6) (Figure 1B). Accordingly, statistical analyses revealed a positive correlation between CELF2+, OLIG2+, and Ki67+ in GBM cells (R = 0.8, *p* < 10^−4^) (Figure 1C). On the other hand, CELF2 expression is inversely correlated with that of miR-199a-3p (Figure 1C), which is expressed in differentiated GBM cells and in OLIG2-negative, low-mitotic index tumor territories [13]. Triple co-immunolabeling performed in GBM tissues from two different patients showed cells that co-expressed CELF2, OLIG2, and CCNA (a marker of dividing cells) (Figure 1D). Using the single-cell RNAseq dataset published by Neftel et al. [9], we confirmed that CELF2 mRNA expression was detected in each of the four cell subtypes, OPC-like, NPC-like, AC-like, and MES-like. However, the highest proportion of cells expressing CELF2—and the highest mean expression were observed in the OPC-like population subtype, which also contained the highest proportion of OLIG2-expressing cells (Appendix A). These results show that CELF2 is expressed not only in OLIG2-positive proliferating cells, but also in all GBM cell subtypes.

We then analyzed the prognostic value of CELF2 for predicting GBM aggressiveness by querying the TCGA database (http://gliovis.bioinfo.cnio.es, accessed on 10 August 2023). While CELF2 expression is a marker of good prognosis in low-grade glioma (Appendix A), we found conversely that CELF2 had no prognostic value in GBMs (Figure 1E), with the exception of classical non-methylated subtypes on the promoter of MGMT, where high CELF2 expression was of poor prognosis (Figure 1F). Taken together, these results indicate that GBM cells not expressing CELF2 and GBM cells expressing it strongly coexist in the same tumor. Furthermore, CELF2 expression in GBMs is associated with an OLIG2-positive proliferative cell phenotype.

### 3.2. CELF2 Is Expressed in GSCs and Is Downregulated upon Cell Differentiation

We then investigated CELF2 expression in previously characterized PDCs derived from three separate patients [8,11,12,13,26] either maintained under a GSCs phenotype or induced to differentiate for 4 days using a medium containing 1% serum. High levels of CELF2 protein were detected via immunoblotting in self-renewing GSCs, while decreasing progressively over time during serum-induced differentiation (Figure 2A,B). We previously demonstrated that forcing miR-199a-3p expression in GSCs resulted in a repressed stemness marker expression (Figure 2D) [13] and aggressiveness in vivo [13]. Interestingly, the transfection of miR-199a-3p into GSCs resulted in a reduced CELF2 expression (Figure 2C) and in the number of CELF2-positive cells (Figure 2D). Conversely, the repression of miR-199a-3p expression in differentiated PDCs using a specific anti-miR sequence prevented the CELF2 repression in differentiated cells (Figure 2E). Consistent with the results described in Figure 1, these results show that CELF2 protein is expressed in proliferating GSCs, whereas it is downregulated in more differentiated cells. These results not only confirm the high expression of CELF2 in proliferating GSCs, but also indicate that this protein can be regulated upon phenotypic change.

### 3.3. CELF2 Is Required for GSC Maintenance and Tumorigenic Potential

To determine the biological role of CELF2 in GSCs, we stably reduced its expression in GB5, GB1, and TG6 using CELF2-specific shRNA (shCELF2). CELF2 inhibition in GSCs was assessed via Western blot and immunofluorescence (Figure 3A and Appendix A). The effect of CELF2 invalidation on the expression of GSC markers, such as OLIG2, NANOG, NESTIN, and SOX1, was assessed via immunofluorescence in shCELF2 cells and quantified against the same cells transduced with an irrelevant control shRNA sequence (shCTL). CELF2 knockdown led to a significant decrease in the expression of OLIG2 as well as NANOG, NESTIN, and SOX1 (Figure 3A–C, Appendix A). In addition, PDCs lacking CELF2 lost their clonal property as evidenced by the repression of their ability to generate primary, secondary, and tertiary spheroids after single-cell seeding (Figure 3D). These results were confirmed using the ELDA method [28] (Appendix A). As a result, the number of CCNA-positive cells, which was above 15% in GSC-shCTL, dropped to values below 10% in GSCs with silenced CELF2 expression (Figure 3E) while the rescue of CELF2 expression restored the number of proliferating cells (Figure 3E, right histogram). Furthermore, CELF2 repression resulted in the loss of the ability of GSCs to form 3D spheroids in mass culture, in favor of the formation of flat-cell aggregates (Figure 3F). We next assessed the effect of CELF2 knockdown on the tumor-initiating properties of GSCs in an orthotopic xenograft model in nude mice. GB5-shCELF2 and GB1-shCELF2 cells, as well as their -shCTL counterparts, were then grafted into the brain of nude mice (*n* = 5 for each group). Tumor growth, monitored using in vivo bioluminescence imaging, was detected in all mice grafted with GB5-shCTL or GB1-shCTL, whereas no signal was detected in mice grafted with cells depleted for CELF2 expression (shCELF2) even three to five weeks after transplantation (Figure 3G, left and middle panels). As a result, all mice bearing GSC-shCTL cells had to be euthanized 3–7 weeks after transplantation, whereas all mice grafted with CELF2-depleted cells survived for 10 weeks and until the experiment was stopped (Figure 3G, right panel). Collectively, these results demonstrate that CELF2 is a cornerstone of GBM-cell aggressiveness, stimulating a proliferative stem-like phenotype.

### 3.4. CELF2 Controls the Level and Distribution of H3K9me3 in Proliferating GSCs

The fact that CELF2 depletion is sufficient to induce the loss of a stem-like and aggressive phenotype in GBM cells led us to determine its involvement in major epigenetic events controlling cellular functions such as chromatin remodeling. The level of histone H3 methylation on lysine residues K4, K3, and K27 was therefore assessed by immunostaining in GSCs expressing CELF2 or not. The expression of histone markers H3K4me1, H3K4me2, and H3K4me3 related to open chromatin and gene transcription did not change upon CELF2 invalidation (Figure 4A). In contrast, we observed a reduction in all H3 marks associated with heterochromatization, i.e., H3K9me2, H3K9me3, H3K27me2, and H3K27me3, with the greatest reduction observed for H3K9me3 levels (Figure 4A,B; Appendix A). This reduction in H3K9me3 expression was confirmed via Western blot (Figure 4C). This observation appears to be a direct consequence of the loss of CELF2 rather than a change in cell fate, since the transient depletion of CELF2 with siRNA leads to the same loss of H3K9me3 levels (Figure 4D). Consistently, blocking the activity of histone methyl transferase G9a (also known as EHMT2) involved in the methylation of H3K9 and H3K27 with a specific chemical inhibitor (G9a-IN-1) reproduced the decrease in K9 and K27 methylation profiles obtained upon CELF2 invalidation, with the strongest repression occurring for H3K9me3 (Figure 4E). Interestingly, the G9a inhibitor strongly repressed GSC clonal proliferation (Figure 4F), suggesting the contribution of heterochromatization in controlling GSC self-renewal. Based on this result, we determined the expression of G9a and of tripartite motif containing 28 (TRIM28, KAP1) genes, both involved in the methylation of the repressive H3K9 and H3K27 markers via the TRIM28-metidated recruitment of G9a. We first showed the decrease in each gene at mRNA level upon CELF2 inhibition (Figure 4G). To determine whether CELF2 is bound to the messenger RNAs of these two genes, we performed RNA immunoprecipitation (RIP) experiments using a CELF2-specific antibody. We then performed qRT-PCR using primers, targeting the regions of TRIM28 and G9a mRNA, previously shown to be bound by CELF2 in other cells (Appendix A). RIP assays revealed a strong enrichment of CELF2 binding on several regions of TRIM28 and G9a mRNAs in control GSCs, whereas this enrichment was lost in CELF2-deficient GSCs (Figure 4H). Western blot analysis also revealed that TRIM28 and G9a protein levels decreased in the absence of CELF2 as did the number of TRIM28-positive cells (Figure 4I,J). Taken together, these results show that CELF2 interacts directly with G9a and TRIM28 mRNAs and controls the expression level of these two genes. This CELF2-dependent regulation is presumably mandatory to maintain appropriate levels of methylated forms of H3K9 and H3K27 in proliferating GBM cells.

### 3.5. A CELF2-Dependent Model of H3K9me3 Distribution in the GSCs’ Chromatin Landscape

As H3K9me3 marks were most affected by CELF2 inhibition, we then compared their distribution in the chromatin of GB5-shCTL cells and their GB5-shCELF2 counterparts, by ChIP-seq analysis using H3K9me3-specific antibody. The results showed 4661 genomic regions whose association with H3K9me3 marks was significantly enriched or decreased upon CELF2 invalidation. Most of these regions (3116, 66.8%) were depleted in H3K9me3 upon CELF2 invalidation, while 1545 regions (33.1%) were enriched (Figure 5A), showing that CELF2 controls both the level and the distribution of H3K9me3 in the chromatin landscape. The analysis showed that a total of 4846 genes were associated with these genomic regions and that the localization of their transcription start site (TSS) to the H3K9me3 recruitment site could extend up to several hundred thousand bp (Figure 5B). Among these, we focused on the 1182 genes whose TSS are localized at a distance of −10 kb to 10 kb of H3K9me3 recruitment sites (Figure 5B). We then cross-referenced this list of 1182 genes with RNAseq data performed in the same cells invalidated or not for CELF2 expression. As shown in Figure 5C, CELF2 repression impacted over 50% of the transcriptome, with a total of 7966 genes regulated, 28% and 25% down- or up-regulated, respectively. The results of the cross-analysis identified a short list of 104 genes that have recorded variations in H3K9me3 recruitment to their proximal regulatory sequences and that are effectively regulated upon CELF2 invalidation (Figure 5D, Appendix A). Ontological analysis of the genes’ molecular functions revealed three main functional groups linked to transcription regulator, binding, and catalytic activities corresponding to 15%, 33%, and 14% of the 104 genes, respectively (Figure 5E). The identification of functional and physical interactions between the proteins encoded by these genes using the STRING-DB algorithm (https://string-db.org, URL accessed on 22 July 2023) revealed a major interaction network, with SOX3 being an important node of this network (Figure 5F), suggesting a central regulatory role in GSCs.

### 3.6. SOX3 Is a Major Player in the Repression of the OLIG2+ Proliferating Phenotype of GBM Cells

An analysis of H3K9me3 positions at the SOX3 locus in GSCs when CELF2 is expressed (GB5-shCTL) revealed a broad distribution of this histone mark throughout the gene body and its proximal regulatory sequences (Figure 6A, upper panel). This H3K9me3 profile was associated with a very low level of SOX3 gene expression. The invalidation of CELF2 in these same cells (GB5-shCELF2) strongly reduced the H3K9me3 coverage on this gene, while stimulating its expression (Figure 6A, lower panel). The consequence was a strong increase in SOX3 protein level as well as the number of SOX3-positive cells as shown by Western blot and immunofluorescence in GB5 and TG6 invalidated or not for CELF2 expression (Figure 6B,C). To further assess the direct contribution of methylated H3K9 on SOX3 repression, we inhibited the remaining G9a activity residing in GB5-lacking CELF2 (GB5-shCELF2) with the G9a-IN-1 at 15 μM for 2 and 4 days (Figure 6D). Cells treated with G9a-IN-1 showed a robust increase in SOX3 protein levels within 4 days of treatment, which was concomitant with a decrease in H3K9me3 levels (Figure 6D). To determine SOX3 biological function in a CELF2-deficient context, we silenced its expression in GB5-shCELF2 and TG6-shCELF2, using two different sequences of specific siRNA. The invalidation of SOX3 stimulated the proliferation and mitotic markers CCNA and H3ser10-p, as well as OLIG2 expression (Figure 6E–H). It is noteworthy that the rescue of CELF2 restored both CCNA and OLIG2 expression while repressing SOX3 (Figure 6I). These results demonstrate that the CELF2-mediated repression of SOX3 in patient-derived GSCs is crucial for maintaining a mitotic/OLIG2-positive cell phenotype.

We then assessed the respective expression of CELF2 and SOX3 in human GBM tissue samples. An analysis of RNAseq studies by Neftel et al. [9] showed that in contrast to CELF2 and OLIG2-positive cells, which were abundant in OPC-like subtypes (80.34% +/− 7.18 and 82.18% +/− 6.5, respectively), SOX3-positive cells represented only 11.6% +/− 1.42 of the cell population (Appendix A). Analyses performed on three different paraffin-embedded samples from GBM patients revealed that SOX3 and CELF2 were mutually exclusive (Figure 7A–C). Similarly, only rare cells co-expressing SOX3 and KI67 were detected, corresponding to less than 0.08% of the total number of SOX3+ cells (Figure 7A–C). These results indicate that SOX3-positive cells do not divide and are not associated with CELF2 expression in the tumor. An analysis of patient survival from the TCGA, CGGA, and Rembrandt GBM datasets showed that patients expressing SOX3 at a level above the median expression appeared to have better survival (Figure 7D–F). Taken together, these results demonstrate that by orchestrating the epigenetic repression of SOX3, CELF2 prevents the indolent behavior of GBM cells in favor of a mitotic and OLIG2-positive phenotype.

## 4. Discussion

In this study, we sought to better understand the expression and biological function of CELF2 in the context of the functional hierarchy that resides in GBMs. Using a collection of human GBM samples as well as patient-derived GSC-enriched spheroid cultures C, we reveal that CELF2 can be highly expressed in proliferating the OLIG2-positive GBM cell population, whereas it is not expressed in other tumor cells, illustrating the complexity of intra-tumor heterogeneity in GBMs. We also found that CELF2 is necessary and instrumental for maintaining aggressiveness and stem-like features in proliferating GSCs. The depletion of CELF2 in GSCs leads to the repression of stemness markers, including OLIG2, the loss of clonal proliferation, as well as the loss of the cellular capacity to form and develop tumors in nude mice brains. We therefore reveal an oncogenic role for CELF2, which promotes the stemness and proliferation of GBM cells. This oncogenic function of CELF2 contrasts with its repressive effect on cell proliferation previously reported in various cancers such as breast cancer, non-small-cell lung carcinoma or even glioma, where CELF2 is underexpressed [20,21,22,23,24]. In these cancers, CELF2 expression is down-regulated via promoter methylation [21] or miRNA targeting [23,24,25], which is probably not the case in all aggressive tumor cells. In the context of the functional hierarchy that resides in tumors, CELF2 expression and functions appear to depend on tumor cell phenotypes, since oncogenic or tumor suppressive activities have been demonstrated in patient-derived cancer stem cells, or, conversely, in tumor cell lines devoid of cancer stem cells, respectively. The importance of cancer stem cell phenotype in determining the biological function of CELF2 is further underlined by the fact that CELF2 expression is repressed upon exit from stem cell status, promoted by serum or by miR-199a-3p expression, a pro-differentiating miRNA [13]. This functional discordance of CELF2 is exemplified by its good prognostic value in low-grade gliomas, whereas it is of poor prognosis in classical GBM subtypes not methylated on the MGMT promoter.

Based on its cellular localization, CELF2 is a regulator of nuclear and cytoplasmic RNA processing events such as alternative splicing and translation [17]. Although best known as a regulator of alternative splicing, the involvement of CELF2 as well as CELF1, another member of the CUG-BP family, has previously been reported in the regulation of translation and/or mRNA stability [17]. Accordingly, we found that CELF2 efficiently binds to TRIM28 and G9a mRNAs in GSCs. The expression of CELF2 in this context was mandatory to enable appropriate expression levels of mRNA and the resulting proteins. The TRIM28 protein is involved in chromatin silencing through its interaction with multiple transcription factors and epigenetic modifiers such as histone methyltransferases like G9a [30]. G9a is a histone–lysine N-methyltransferase (KMT) well known for catalyzing mono- and di-methylated states of histone H3 at lysine residues 9 and 27 [30,31]. While H3K27me2, H3K27me3, and H3K9me2 levels were altered via G9a inhibition in patient-derived GSCs, H3K9me3 was most affected. This confirms that G9a may also be involved in the production of the tri-methylated form of H3K9me3 [32,33]. Interestingly, it is well established that G9a is overexpressed in cancer and contributes to the epigenetic silencing of tumor suppressor genes, thus constituting a marker of poor prognosis [34]. It is also involved in the inhibition of the differentiation of embryonic stem cells (ESCs) into neural precursor cells (NPCs) [32]. These important biological functions of G9a are consistent with the function of CELF2 which we have identified in GSCs. The invalidation of CELF2 not only mimicked the repressive effect of G9a inhibition on H3K9me3 levels, but also altered the distribution of this repressive mark in the chromatin landscape of GBM cells. We identified a short list of 104 genes whose expression is altered in the absence of CELF2, while their regulatory sequences spanning 10,000 kb around the TSS are associated with H3K9me3 marks. Among these genes, we focused on the HMG-box transcription factor SOX3, a member of the SOXB1 family comprising SOX1 and SOX2. In normal NPCs, SOX1, SOX2, and SOX3 are co-expressed and contribute to maintaining neural cells in an undifferentiated state [35,36]. This is in agreement with studies that have described SOX3, as well as SOX2, as players involved in the maintenance of human ESC identity although SOX3 has been shown to be unable to compensate for the lack of SOX2 to rescue ESC self-renewal [37]. Functionally, in the context of astrocyte differentiation, a recent study in normal neuronal tissue revealed that genes involved in glial commitment are preselected by SOX3 binding to their regulatory sequences, resulting in cell specification toward the astrocytic lineage [38]. This finding proves that SOX3 is essential for astrocyte differentiation. In PDCs, we unexpectedly discovered that, in contrast to normal NPCs and human ESCs, SOX3 expression is inversely correlated with that of other SOXB1 family members as well as stemness and aggressive markers such as NESTIN, NANOG, and OLIG2. Here, we have shown that SOX3 has an anti-proliferation and anti-stemness role in PDCs, which contrasts with the positive contribution to cancer cell proliferation and tumor progression previously described in various cancers, including gliomas [39,40]. Indeed, SOX3 effectively repressed CCNA and OLIG2 protein expressions. This observation is reinforced by the SOX3 protein expression profile in wild-type IDH human GBM samples, where CELF2 and SOX3 expressions are mutually exclusive and reveal the presence of a population of SOX3-positive non-mitotic tumor cells that are also negative for CELF2, OLIG2, and KI67 expressions. These SOX3-positive cells are associated with an indolent tumor phenotype observed in PDC and GBM tissue samples. In line with these results, patients expressing higher levels of SOX3 have better survival.

## 5. Conclusions

We reveal for the first time an oncogenic function of CELF2 that promotes a proliferative and OLIG2-positive phenotype in GSCs. In this context, CELF2 acts as a master epigenetic factor, determining the level and distribution of H3K9me3 in the chromatin landscape of GSCs, through the regulation of TRIM28 and G9a expression. Through this mode of action, CELF2 can repress genes such as SOX3, which we have shown to oppose the phenotype of mitotic/OLIG2-positive tumor cells. Our study provides new insights into the mechanisms responsible for GBM malignancy that could lead to the development of new anti-tumor therapies directed against the glioblastoma stem cell phenotype.

## Figures and Tables

**Figure 1 cancers-15-05038-f001:**
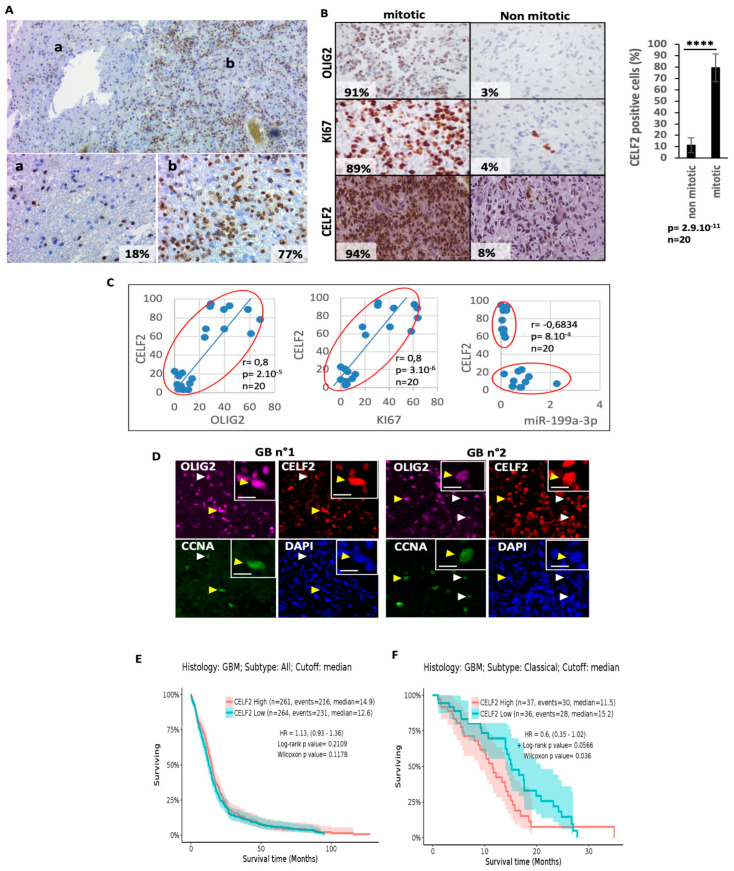
The RNA binding protein CELF2 is a signature of aggressiveness and mitotic OLIG2-positive cells. (**A**) Immunohistochemistry (IHC) showing specific CELF2 staining in zones of low (**a**) or high (**b**) cellular density of a human GBM sample. (**B**) CELF2, KI67, and OLIG2 expression profile in typical mitotic and non-mitotic tumor territories (*n* = 20) revealed via IHC staining. Histogram shows the number of CELF2-positive cells in non-mitotic and mitotic tumor territories, respectively (*n* = 20 *p* value < 0.0001 ****). (**C**) Correlation between the number of CELF2-positive cells and OLIG2, Ki67, and miR-199a-3p in non-mitotic and mitotic GBM territories (*n* = 20). Correlations have been quantified using a Pearson correlation test. (**D**) Co-immunostaining showing the co-expression of CELF2 with CCNA and OLIG2 in two different human GBM samples. Arrows show examples of cells co-expressing CCNA, OLIG2, or CELF2 staining. The yellow arrow points to the cell that has been enlarged in the box (right corner of the panel). The scale = 10 µM. (**E**,**F**) Survival analysis showing the CELF2 prognosis value in GBM (**E**) and in GBM classical subtype unmethylated on MGMT promoter (**F**). The analysis was performed using the Gliovis interface (http://gliovis.bioinfo.cnio.es, accessed on 10 August 2023).

**Figure 2 cancers-15-05038-f002:**
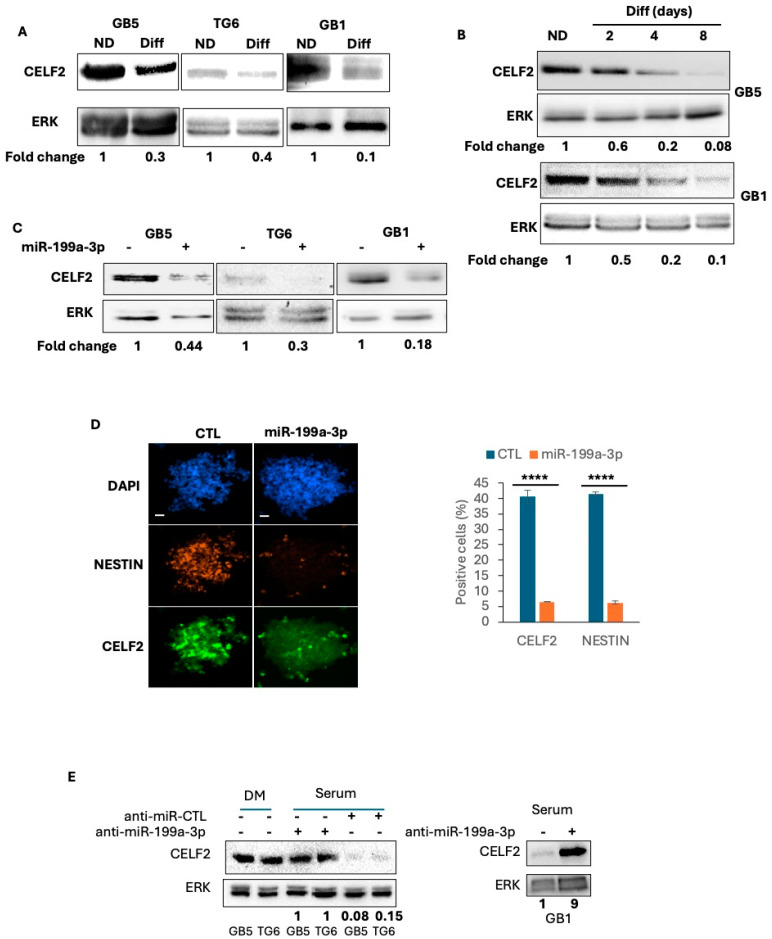
CELF2 expression and regulation in patient-derived GSCs. (**A**) Western blot analysis showing CELF2 expression in proliferating GSCs and their serum-differentiated counterparts (three days of differentiation). (**B**) Western blot showing CELF2 expression during the time course of GSC differentiation in serum medium. (**C**) GB5, TG6 and GB1 have been transfected by a synthetic miR-199a-3p or by a control non-relevant miRNA. CELF2 expression is revealed in both condition by western blot. (**D**) Immunofluorescence showing the expression of CELF2 or Nestin (marker of GSCs) in GB5 stably expressing a synthetic miR-199a-3p or a control non-relevant sequence. Scale = 100 µM. Histogram on the right displays the quantification of the number of positive cells (*p* value < 0.0001 ****). (**E**) GB5, TG6 and GB1 have been transfected by an anti-miR-199a-3p (synthetic sequence) or by a non-relevant sequence as anti-miRNA control. CELF2 expression is revealed by western blot.

**Figure 3 cancers-15-05038-f003:**
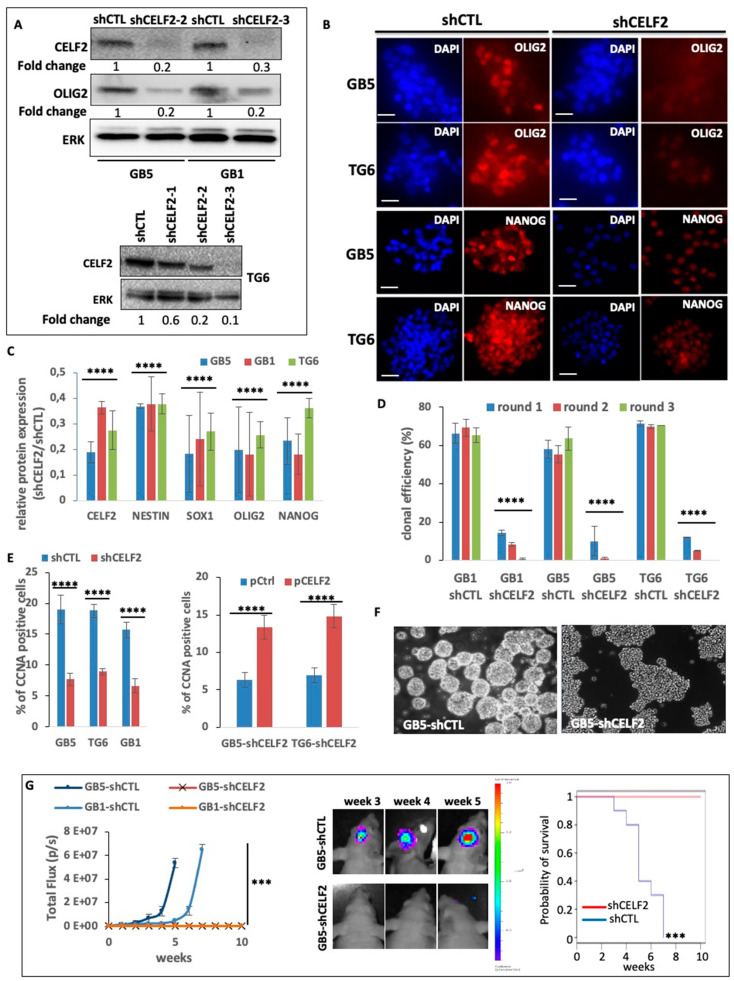
CELF2 promotes stem-like features and aggressive phenotype in GBM cells. (**A**) Protein extracts from GB5, GB1, and TG6 deficient (shCELF2) or not (shCTL) for CELF2 expression are used for Western blot analysis. The membranes were immunoblotted with CELF2, OLIG2 and total ERK (loading control) specific antibodies. (**B**) Immunofluorescence showing OLIG2 and NANOG expression in GB5 and TG6 deficient (shCELF2) or not (shCTL) for CELF2 expression. Scale = 50 µM. (**C**) Histogram showing the expression ratio of CELF2, OLIG2, NESTIN, NANOG, and SOX1 (fold change) between GB5, GB1, and TG6 deficient (shCELF2) or not (shCTL) for CELF2 expression, respectively. The values were obtained from immunofluorescence image acquisitions and fluorescence quantification (NIS-elements software, Nikon). Error bars are the meaning of three independent experiments (*p* value < 0.001 ****). (**D**) Evaluation of clonal capacity. Three successive rounds of clonal amplification assay have been performed using GB1, GB5, TG6 cells deficient (shCELF2) or not (shCTL) for CELF2 and seeded at a density of 1 cell/well in 96-wells plates (*p* value < 0.001 ****). (**E**) (**Left histogram**) Quantification of immunofluorescence showing the number of CCNA-positive cells, in GB5, GB1, and TG6 deficient (shCELF2) or not (shCTL) for CELF2 expression (*p* value < 0.001 ****). (**Right histogram**) Quantification of immunofluorescence showing the number of CCNA-positive cells, in GB5 and TG6 deficient (shCELF2) for CELF2 expression and transfected either with a CELF2 expressing vector or a control expression plasmid (*p* value < 0.001 ****). (**F**) Phase contrast images showing that GB5 deficient for CELF2 expression have lost their capacity to form spheroids in 3D culture. (**G**) Luminescent GB5 and GB1 spheroids deficient (shCELF2) or not (shCTL) for CELF2 expression have been engrafted in the brain of nude mice (*n* = 5/culture cell) for several weeks. Tumor growth has been assessed and quantified via live imaging (IVIS lumina III). (**Left**) The results have been plotted according to the time following tumor initiation (weeks) and show a significant difference between cells deficient or not for CELF2 (Wilcoxon Mann–Whitney test, *p* value < 0.01 ***). (**Middle**) Pictures (live imaging) exemplifying the difference in tumor initiation and growth between shCTL versus shCELF2. (**Right**) Kaplan–Meier (log-rank test) analysis of mice survival. The group of mice xenografted by GB1 and GB5 shCTL has been compared to the group of mice xenografted by GB1 and GB5 shCELF2 (significative difference *p* < 0.01). The analysis has been performed using the BIOSTATGV platform (https://biostatgv.sentiweb.fr/?module=tests/surv, accessed on 10 August 2023.).

**Figure 4 cancers-15-05038-f004:**
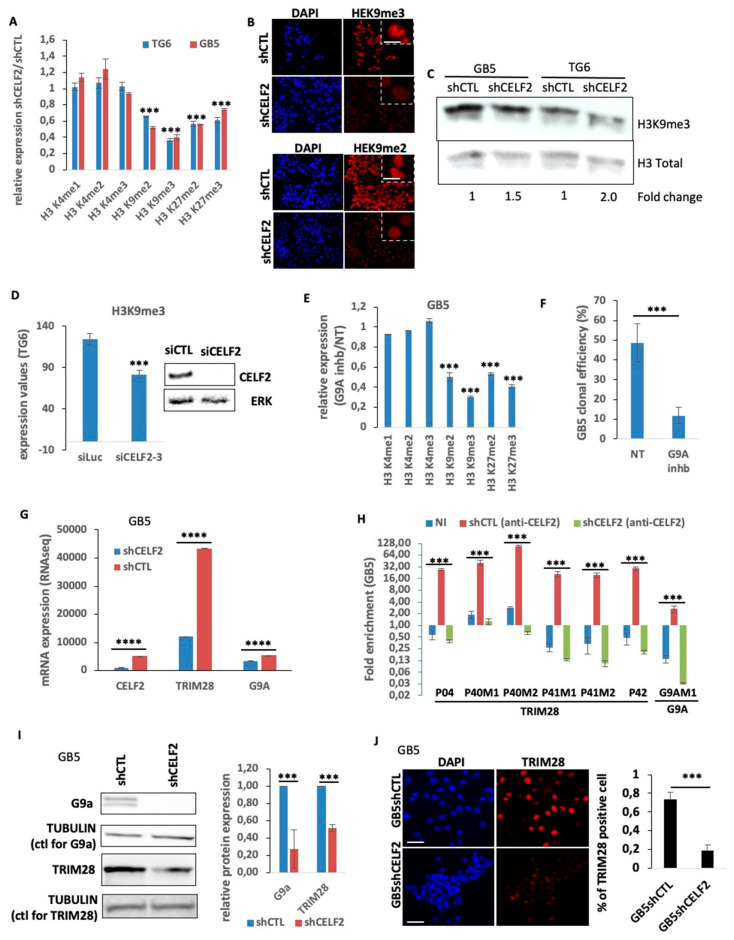
CELF2 interacts with TRIM28 and G9A mRNA, regulates their expression, and controls the methylation level of H3K9. (**A**) Quantification of immunofluorescence showing the ratio of the expression of various histone marks in TG6 and GB5 deficient (shCELF2) or not (shCTL) for CELF2 expression (*p* value < 0.001 ***). (**B**) Immunofluorescence showing H3K9me3 level in GB5 deficient (shCELF2) or not (shCTL) for CELF2 expression. Scale = 10 µM. (**C**) Western blot performed in GB5 and TG6, showing H3K9me3 level in cells invalidated (shCELF2) or not (shCTL) for CELF2 expression. (**D**) TG6 have been transiently transfected with a CELF2-specific (siCELF2) or a control siRNA (siLuc). A total of 48 h later, the cells were subjected to immunofluorescence to reveal H3K9me3 expression level. The histogram shows the immunofluorescence quantification (*p* value < 0.001 ***). The Western blot shows the siRNA efficiency. (**E**) Quantification of immunofluorescence performed in GB5 treated or not with G9a-IN-1, showing the ratio of expression of various histone marks (*p* value < 0.001 ***). (**F**) GB5 cells have been treated by G9a-IN-1 and seeded at a density of 1 cell/well in 96-wells plates for spheroid formation assay. Three independent plates have been then quantified (*p* value < 0.001 ***). (**G**) RNAseq results, showing the difference of CELF2, EHMT2 (G9a), and TRIM28 mRNA levels in GB5shCELF2 and GB5shCTL (*p* value < 0.0001 ****). (**H**) RNA precipitation assays using CELF2-specific antibody and followed by QPCR analysis using primers located in TRIM28 and G9a mRNA regions previously reported [29] as possibly bound by CELF2 (*p* value < 0.001 ***). (**I**) Western blots performed with protein extract from GB5 deficient (shCELF2) or not (shCTL) for CELF2 expression, showing the difference of EHMT2 (G9a) et TRIM28 expression level. Tubulin has been used as loading control. Right panel, Western blot quantification using Image J software (*p* value < 0.001 ***). (**J**) Immunofluorescence performed in GB5-shCTL and GB5-shCELF2, showing the number of TRIM28-positive cells. Scale = 100 µM. The histogram shows the immunofluorescence quantification (*p* value < 0.001 ***).

**Figure 5 cancers-15-05038-f005:**
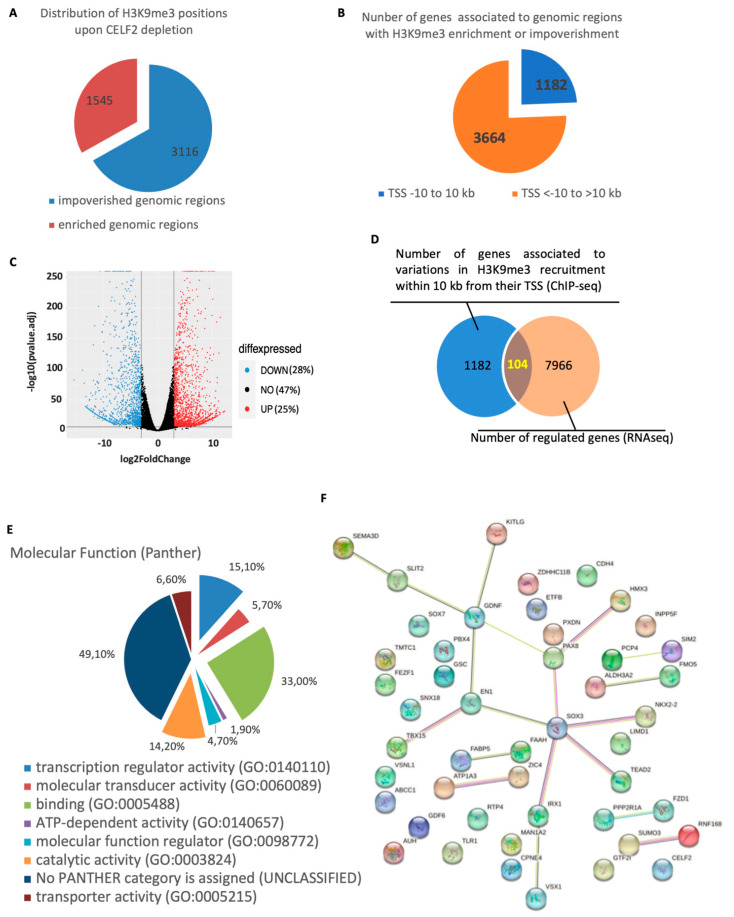
CELF2 controls the level and the distribution of H3K9me3 marks in patient-derived GSCs chromatin landscape. (**A**) ChIP-seq analysis performed with H3K9me3-specific antibodies using cross-linked chromatin extracted from GB5-shCTL or GB5-shCELF2 cells. Pie chart showing the number of genomic regions that are impoverished or enriched in H3K9me3 recruitment upon CELF2 invalidation. (**B**) Pie chart showing the number of genes associated with the genomic regions described in (**A**) and displayed according to the distance from H3K9me3 recruitment site. (**C**) RNAseq analysis has been performed to compare GB5-shCELF2 and GB5-shCTL transcriptomes. Volcano plot revealing that CELF2 controls the regulation of more than 50% of gene expression. (**D**) Venn diagram showing the number of genes (104) that are both up- or down-regulated and whose proximal regulatory sequences were impoverished or enriched in H3K9me3 recruitment when CELF2 is depleted. (**E**) Pie chart showing the result of a functional analysis of the 104 genes identified by the cross-analysis between ChIP-seq and RNAseq, using the Gene Ontology Resource (http://geneontology.org UR accessed on 22 July 2023). (**F**) Network interactome developed by the genes belonging to the transcription regulator, binding, and catalytic activity identified in the functional analysis.

**Figure 6 cancers-15-05038-f006:**
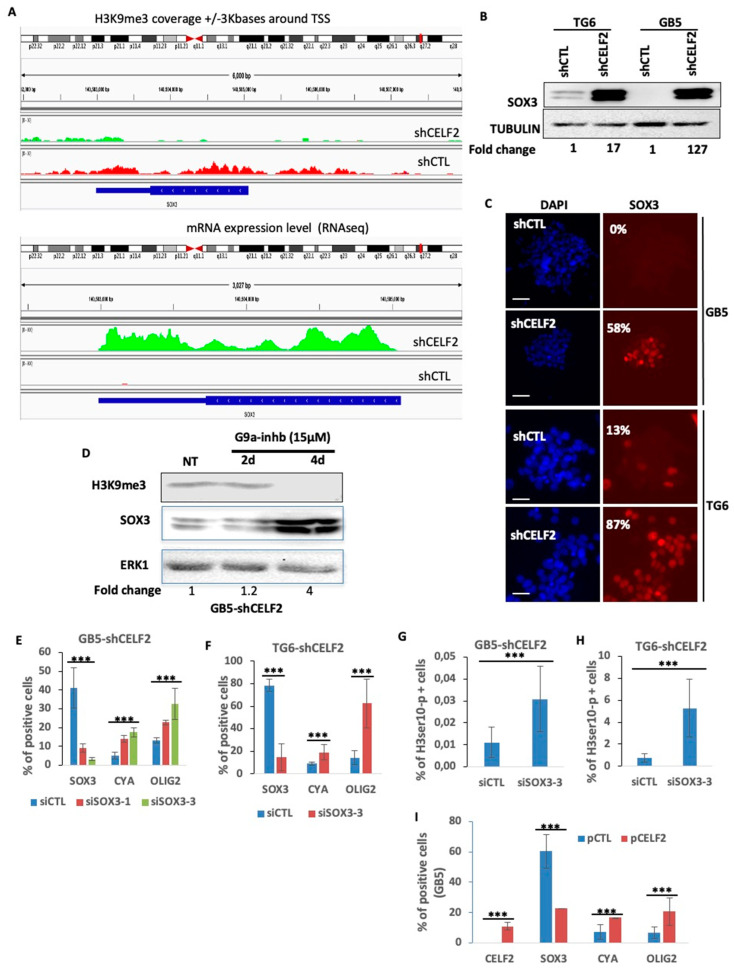
SOX3 is a major repressor of OLIG2 and CCNA expression in CELF2-deficient GSCs. (**A**) Graphic comparing the difference of H3K9me3 recruitment on SOX3 gene and proximal promoter in GB5-shCTL and GB5-shCELF2 (**upper panel**). As well as the difference in the number of reads (RNAseq) detected for SOX3 mRNA in GB5-shCTL and GB5-shCELF2 (**lower panel**). (**B**,**C**) SOX3 protein expression revealed through Western blot (**B**) or immunofluorescence (the values represent the average of positive cells (%) from three separate experiments, scale = 100 µM) in GB5 and TG6 cells deficient (shCELF2) or not (shCTL) for CELF2 expression. (**D**) Western blot showing the level of SOX3 and H3K9me3 expression performed in GB5 deficient for CELF2 expression (GB5-shCELF2) treated or not with the G9a-IN-1 (15 μM) during two and four days. ERK was used as loading control. (**E**–**G**) Quantification of immunofluorescences showing the number of SOX3, CCNA, OLIG2-positive cells or (**H**) H3ser10-p in GB5-shCELF2 (**left panel**) or TG6-shCELF2 (**right panel**), transfected with a control siRNA (siCTL) or two different sequences of SOX3-specific siRNA (siSOX3-1 or siSOX3-3) (*p* value < 0.01). (**I**) Rescue of CELF2 expression. Quantification of immunofluorescence showing the number of CELF2, SOX3, CCNA, OLIG2-positive cells in GB5-shCELF2 transfected either with a control vector (pCTL) or a CELF2 expression vector (pCELF2) (*p* value < 0.001 ***).

**Figure 7 cancers-15-05038-f007:**
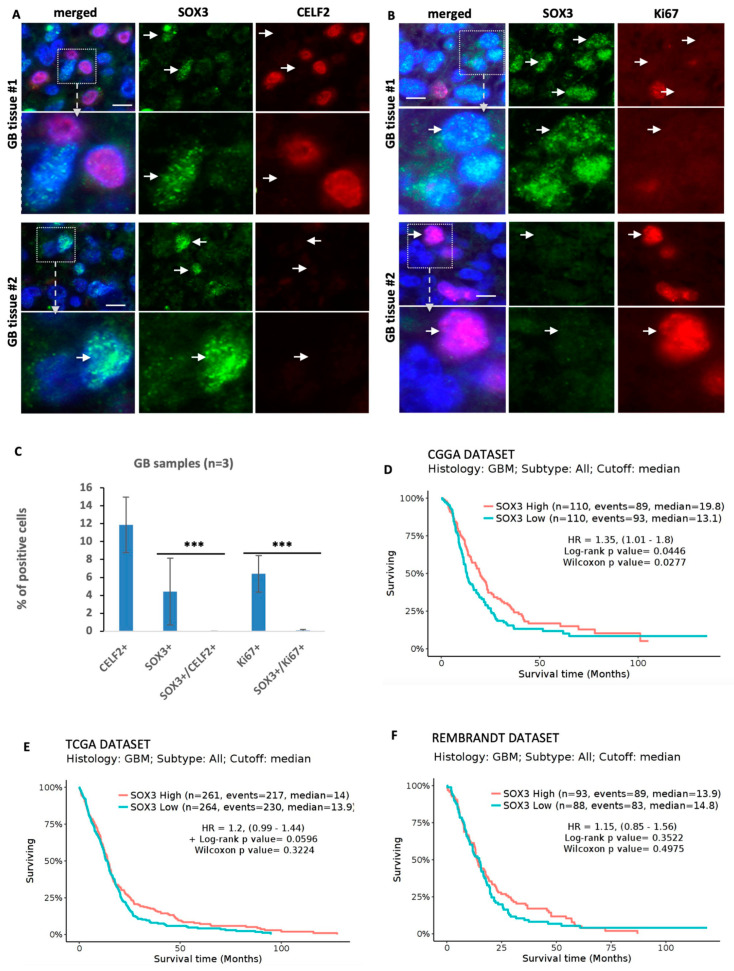
SOX3 expression in GBM correlates with non-mitotic and CELF2-negative cells. (**A**) Co-immunostaining showing SOX3 and CELF2 expression or (**B**) SOX3 and Ki67 expression in two different human GBM samples. White arrows show some positions of SOX3-positive cells in the CELF2 and ki67 panels. Scale = 10 µM. (**C**) Quantification of the number of CELF2-, SOX3-, and Ki67-positive cells and the number of SOX3/CELF2 and SOX3/Ki67-double-positive cells in 3 different human GBM samples. (Wilcoxon Mann–Whitney test, *p* value < 0.001 ***). (**D**–**F**) Patient survival analysis comparing two groups of patients either expressing SOX3 above (red curve) or below (blue curve) the median of expression. This analysis has been performed using GLIOVIS website using the CGGA (**D**), TCGA (**E**), or the Rembrandt (**F**) datasets (http://gliovis.bioinfo.cnio.es, accessed on 22 July 2023).

## Data Availability

The data that support the findings of this study are available from the corresponding author upon reasonable request.

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
