# Peer review of "CELF2 Sustains a Proliferating/OLIG2+ Glioblastoma Cell Phenotype via the Epigenetic Repression of SOX3"

_cancers, 2023, doi:10.3390/cancers15205038_

Round 1

Reviewer 1 Report

The manuscript explores the role of CELF2, an RNA-binding protein, in glioblastomas (GB), aggressive brain tumors with stem-like tumor cells that contribute to treatment resistance and tumor recurrence. The authors provide evidence that CELF2 is expressed in mitotic and OLIG2 positive GB cells and is involved in maintaining the proliferative/OLIG2 cell phenotype, thus promoting tumor growth. Additionally, the study demonstrates that CELF2 deficiency reduces tumor growth in mice. The authors highlight the relationship between CELF2 and the expression of TRIM28 and G9a, which leads to an epigenetic profile associated with the silencing of the SOX3 gene. They also find that higher expression of SOX3 correlates with longer life expectancy in patients, indicating the clinical relevance of these findings for novel anticancer strategies against GB.

While the manuscript presents interesting findings and supports them with experimental data, several concerns need to be addressed before considering it for publication:

The manuscript should address the issue of GB heterogeneity, as it plays a significant role in GB treatment outcomes. The authors mention a negative correlation between CELF2 expression and patient survival specifically in the classical subtype of GB, but they fail to provide information about the GB cell lines used in the experimental setting. Clarifying the subtypes of these cell lines and relating them to clinical observations would enhance the manuscript's context.

As the manuscript does not focus on low-grade IDH-mt glioma, it is recommended to remove Figure 1E, which may not directly contribute to the main objectives of the study.

To evaluate stemness in the absence of CELF2, the authors should employ the classical Extreme Limiting Dilution Assay (ELDA) and consider rescuing the deficiency through an overexpression system. These additional experiments would provide a more comprehensive understanding of the role of CELF2 in GB stemness.

The authors should support their claim regarding CELF2-mediated regulation of H3K9 methylation by employing quantitative methods such as immunoblot analysis in multiple GBM cell lines with different subtypes. Addressing the heterogeneity of GBM through such experiments would strengthen their conclusions.

The use of different datasets and patient cohorts to evaluate the clinical significance of SOX3 in Figure 6D-E requires clarification. It is recommended to focus solely on high-grade GBM, remove the low-grade glioma data, and validate the findings using at least three independent datasets.

Addressing these concerns would enhance the manuscript's scientific rigor and ensure that it meets the criteria for publication.

Need minor revision. 

Author Response

While the manuscript presents interesting findings and supports them with experimental data, several concerns need to be addressed before considering it for publication:

Thank you for helping us to improve our manuscript with all your fruitful questions. You will find below a point by point response to your comments.

The manuscript should address the issue of GB heterogeneity, as it plays a significant role in GB treatment outcomes. The authors mention a negative correlation between CELF2 expression and patient survival specifically in the classical subtype of GB, but they fail to provide information about the GB cell lines used in the experimental setting. Clarifying the subtypes of these cell lines and relating them to clinical observations would enhance the manuscript's context.

Details about the patient derived GSC have been added in the materials and methods section. The following sentences: GB5 and TG6 display a gain on chromosome 7, without the loss of PTEN while GB1 displays a gain on chromosome 7 with the loss of the entire chromosome 10. GB1, GB5 and TG6 display no mutation of IDH1/2 and TP53, active SHH and NOTCH pathways; have been added at line 128.

As the manuscript does not focus on low-grade IDH-mt glioma, it is recommended to remove Figure 1E, which may not directly contribute to the main objectives of the study.

The panel E of the figure 1 has been removed and placed in supplemental figures as figure S1.

To evaluate stemness in the absence of CELF2, the authors should employ the classical Extreme Limiting Dilution Assay (ELDA) and consider rescuing the deficiency through an overexpression system. These additional experiments would provide a more comprehensive understanding of the role of CELF2 in GB stemness.

The ELDA method has been used with GB5 and TG6 cells. The results obtained confirm the data displayed in figure 3. CELF2 rescue has been performed in GB5 and TG6 (shCELF2) as described in the method section. As displayed in the new figure 3 E, CELF2 re-expression restores CCNA expression in CELF2 deficient GB5 and TG6 (shCELF2).

The following sentence: These results were confirmed by using the ELDA method [28] with GB5 and TG6 cells (supplemental figure S3). Consequently, the number of CCNA-positive cells, which was greater than 15% in GSC-shCTL, significantly dropped to values below 10% in GSCs silenced for CELF2 expression (Figure 3D) while the rescue of CELF2 expression restored the number of cells in proliferation; have been added at line 312.

The authors should support their claim regarding CELF2-mediated regulation of H3K9 methylation by employing quantitative methods such as immunoblot analysis in multiple GBM cell lines with different subtypes. Addressing the heterogeneity of GBM through such experiments would strengthen their conclusions.

A western blot has been performed in GB5 and TG6 cells and displayed in figure 4C and the text has been corrected accordingly at line 342.

The use of different datasets and patient cohorts to evaluate the clinical significance of SOX3 in Figure 6D-E requires clarification. It is recommended to focus solely on high-grade GBM, remove the low-grade glioma data, and validate the findings using at least three independent datasets.

The figure 6D and E show respectively a Kaplan Meier analysis performed with the TCGA and CGGA datasets. In this analysis only GBM were considered. These informations are mentioned in the legend of the figure but to better clarify, we have added the name and the histology type in the title of the graph. In addition, similar analysis has been performed using the Rembrandt dataset, showing identical tendency to an improved survival when SOX3 is expressed above the median of expression, but not statistically significant in this dataset. Accordingly, the text line 417 has been modified as following : Analysis of patient survival from the TCGA, CGGA and Rembrandt GB database showed that patients expressing SOX3 at a level higher than the median expression seemed to benefit from improved survival (Figure 7D-G).

Reviewer 2 Report

In this article, Turchi et al show that CELF2 is associated with proliferating Olig2+ cell type by repressing SOX3.

The article overall is well written, and methods and rationale well explained and worked on.

I have a few major concerns about the work:

1) Using ERK as a loading control for WB is not appropriate. ERK is known to be modulated when GBM cells are differentiated. Infact the authors themselves have shown in ref 13 Almairac et al that p-ERK is elevated upon differentiation. P-ERK and ERK expression levels are interconnected.

2) From methods, the authors appeared to have used 4 different shRNAs for CELF2, but results from only one are presented in the manuscript. Did the other shRNAs not show the same results? It’s hard to base observation based on one shRNA and not be sure this is an off target effect.

3) Figure1E needs labels to day low glade glioma, HGG. etc.. since there is not really a positive association of high CELF2 expression with HGG overall maybe these figures can be moved to supplementary figure 1? I am actually surprised that this is the case given that there was such a strong reduction in stem cells and tumour formation in vivo. Is this because it some sort of off target effect from the single CELF2 shRNA used?

4) Are the cells viable after CELF2 knockdown (KD)? Is there any cell death and no proliferation after KD? Did they stain the aggregates in figure 3F for differentiation markers?

5) With regards to the in vivo experiments presented in Fig 3, this is a remarkable result, were there were any signs of tumours from the H&Es of animals with CELF2 KD? Is there any chance some sort of tumours would form if the animals would have been kept alive for longer?

Minor points:

1) GB should be changed to GBM, it’s the more accepted acronym for glioblastoma.

2) For the differentiation experiment, methods suggest authors use 0.5% serum while results on page 8 says they used 1%.

3) Looking at supplementary fig S1 it doesn’t look like the OPC like cell state shows highest expression of CELF2. How supplementary table 1 shows this is not clear as well. Fig1 D it looks like CELF2 is also expressed on a non Olig2 + cell type suggesting it is not only expression on OPC cell state.

4) General comment on figures- authors need to check that the p-values are listed and that the stars are correctly marked. Some instances there are three stars when there should be one star? Also scale bars required in all images. Missing in some places. Finally in some figures it is not clear what cell line the experiment reflects.

5) Fig 2A –what was the time point if the WB?

6) Fig3G- Add radiance colour scale of IVIS images.

7) Did the chip-seq of CELF2 shRNA confirm the results shown in section 3.4 in figure 4G?

8) There is no WB showing siRNA KD efficiency of CELF2 reduction in figure 4C.

Dear editor,

Thank you for the opportunity to review this article and your patience with me.

This is a well-written manuscript. The work seems novel enough to warrant publication but needs some changes.

Kind regards,

Rochelle

Author Response

The article overall is well written, and methods and rationale well explained and worked on. 

Thank you for all your fruitful questions.

I have a few major concerns about the work:

1) Using ERK as a loading control for WB is not appropriate. ERK is known to be modulated when GBM cells are differentiated. Infact the authors themselves have shown in ref 13 Almairac et al that p-ERK is elevated upon differentiation. P-ERK and ERK expression levels are interconnected.

We agree that it is difficult to identify a good and stable loading control. It tightly depends on the cellular system that one use. In our hands in our patient derived cells, after having performed several dizaine of western blots (cf our numerous publications), it appears that total ERK is not so bad and could be a reliable loading control, as tubulin does, and much more stable that actin, for instance.

2) From methods, the authors appeared to have used 4 different shRNAs for CELF2, but results from only one are presented in the manuscript. Did the other shRNAs not show the same results? It’s hard to base observation based on one shRNA and not be sure this is an off target effect.

Thank you for this important point. As displayed in figure 3A, we have tested three different shRNA. The results show that shCELF2-2 and shCELF2-3 have provided the best results. We used the shCELF2-2 in GB5 and shCELF2-3 in, TG6 and GB1. The effects of CELF2 silencing have therefore been demonstrated using two different CELF2 shRNA sequences. To clarify this, the figure 3A  upper panel has been modified and the choice of the shRNA sequence has been reported in the materials and methods section. Accordingly the text has been modified as following: HSH000802-2 shCELF2-2 has been used to provide shCELF2 GB5. HSH000802-3 shCELF2-3 has been used to provide shCELF2 TG6 and GB1 cells, at the line 149.

3) Figure1E needs labels to day low glade glioma, HGG. etc.. since there is not really a positive association of high CELF2 expression with HGG overall maybe these figures can be moved to supplementary figure 1?

Panel E of the figure 1 concerning low grade glioma has been placed in supplemental figure S2.

I am actually surprised that this is the case given that there was such a strong reduction in stem cells and tumour formation in vivo. Is this because it some sort of off target effect from the single CELF2 shRNA used?

Any off-target effect can be ruled out because, as described in the method section, we used two different sequences of shRNA and that CELF2 expression has been restored resuming CCNA and OLIG2 expression. From my point of view, one explanation can reside in the fact that CELF2 expression is heterogeneous within the same tumor as it coexists CELF2 strongly positive cells among negative cells, in various proportion. This particular finding is hardly depicted by transcriptomic studies, which are the main datasets used for survival analysis.

4) Are the cells viable after CELF2 knockdown (KD)? Is there any cell death and no proliferation after KD? Did they stain the aggregates in figure 3F for differentiation markers?

The cells are viable after CELF2 knock down and the proliferation is repressed as confirmed by the down regulation of CCNA protein. While stemness markers are repressed, the differentiation marker GFAP does not significantly comes up.

5) With regards to the in vivo experiments presented in Fig 3, this is a remarkable result, were there were any signs of tumours from the H&Es of animals with CELF2 KD? Is there any chance some sort of tumours would form if the animals would have been kept alive for longer?

The H&E staining is negative, which is according to the lack of bioluminescent signal. We agree with your comment, we cannot rule out the possibility that rare undetectable cells can become over time resistant to CELF2 knock and be able to form a tumor. However, the goal of this figure was to reveal the striking difference that resides between CELF2 positive and CELF2 invalidated cells in their capacity to grow a tumour, and this discrepancy is clear in this figure.

Minor points:

1) GB should be changed to GBM, it’s the more accepted acronym for glioblastoma.

GB has been changed for GBM

2) For the differentiation experiment, methods suggest authors use 0.5% serum while results on page 8 says they used 1%.

Thank for this comment, the right percentage of serum (1%) has been written in the method section.

3) Looking at supplementary fig S1 it doesn’t look like the OPC like cell state shows highest expression of CELF2. How supplementary table 1 shows this is not clear as well. Fig1 D it looks like CELF2 is also expressed on a non Olig2 + cell type suggesting it is not only expression on OPC cell state.

We totally agree that CELF2 is not necessarily expressed in OLIG2 positive cells, To clarify this point we had in the result section at line 281, the following sentence: These results show that CELF2 is expressed not only in OLIG2-positive proliferating cells but also in every cell subtype of GBM.

4) General comment on figures- authors need to check that the p-values are listed and that the stars are correctly marked. Some instances there are three stars when there should be one star? Also scale bars required in all images. Missing in some places. Finally in some figures it is not clear what cell line the experiment reflects.

Thank you for this comment, these points have been corrected.

5) Fig 2A –what was the time point if the WB?

The time point is three days. This has been added in the legend of the figure.

6) Fig3G- Add radiance colour scale of IVIS images.

The radiance has been added.

7) Did the chip-seq of CELF2 shRNA confirm the results shown in section 3.4 in figure 4G?

G9A and TRIM28 did not came up in the ChIP-seq data set showing that CELF2 mediated regulation of these genes did not occur through H3K9me3. However, the data in figure 4H showing CELF2 binding on TRIM28 and G9A mRNA, strongly suggest a direct post-transcriptionnal regulation.

8) There is no WB showing siRNA KD efficiency of CELF2 reduction in figure 4C.

A western blot showing siCELF2 efficiency has been added to the panel 4D of the new figure 4.

Round 2

Reviewer 1 Report

Authors addressed the majority of the critics, and this revised manuscript is now much improved one.